# Domain Adaptation as a Problem of Inference on Graphical Models

**Kun Zhang**[1*], **Mingming Gong**[2*], **Petar Stojanov**[3]
**Biwei Huang**[1], **Qingsong Liu**[4], **Clark Glymour**[1]
[1] Department of philosophy, Carnegie Mellon University
[2] School of Mathematics and Statistics, University of Melbourne
[3] Computer Science Department, Carnegie Mellon University, [4] Unisound AI Lab
kunz1@cmu.edu, mingming.gong@unimelb.edu.au, liuqingsong@unisound.com
{pstojano, biweih, cg09}@andrew.cmu.edu

## Abstract

This paper is concerned with data-driven unsupervised domain adaptation, where it is unknown in advance how the joint distribution changes across domains, i.e., what factors or modules of the data distribution remain invariant or change across domains. To develop an automated way of domain adaptation with multiple source domains, we propose to use a graphical model as a compact way to encode the change property of the joint distribution, which can be learned from data, and then view domain adaptation as a problem of Bayesian inference on the graphical models. Such a graphical model distinguishes between constant and varied modules of the distribution and specifies the properties of the changes across domains, which serves as prior knowledge of the changing modules for the purpose of deriving the posterior of the target variable $Y$ in the target domain. This provides an end-to-end framework of domain adaptation, in which additional knowledge about how the joint distribution changes, if available, can be directly incorporated to improve the graphical representation. We discuss how causality-based domain adaptation can be put under this umbrella. Experimental results on both synthetic and real data demonstrate the efficacy of the proposed framework for domain adaptation. The code is available at https://github.com/mgong2/DA_Infer.

## 1   Introduction

Over the past decade, various approaches to unsupervised domain adaptation (DA) have been pursued to leverage the source-domain data to make prediction in the new, target domain. In particular, we consider the situation with $n$ source domains in which both the $d$-dimensional feature vector $X$, whose $j$th dimension is denoted by $X_j$, and label $Y$ are given, i.e., we are given $(\mathbf{x}^{(i)}, \mathbf{y}^{(i)}) = (\mathbf{x}_k^{(i)}, y_k^{(i)})_{k=1}^{m_i}$, where $i = 1, ..., n$, and $m_i$ is the sample size of the $i$th source domain. We denote by $x_{jk}^{(i)}$ the value of the $j$th feature of the $k$th data point (example) in the $i$th domain. Our goal is to find the classifier for the target domain, in which only the features $\mathbf{x}^\tau = (\mathbf{x}_k^\tau)_{k=1}^m$ are available. Because the distribution may change across domains, clearly the optimal way of adaptation or transfer depends on what information is shared across domains and how to do the transfer.

In the covariate shift scenario, the distribution of the features, $P(X)$, changes, while the conditional distribution $P(Y|X)$ remains fixed. A common strategy is to reweight examples from the source domain to match the feature distribution in the target domain–an approach extensively studied in

---

machine learning; see e.g., [1, 2, 3, 4]. Another collection of methods learns a domain-invariant feature representation that has identical distributions across the target and source domains [5, 6, 7, 8, 9].

In addition, it has been found that $P(Y|X)$ usually changes across domains, in contrast to the covariate shift setting. For the purpose of explaining and modeling the change in $P(Y|X)$, the problem was studied from a generative perspective [10, 11, 12, 13, 14]–one can make use of the factorization of the joint distribution corresponding to the causal representation and exploit how the factors of the joint distribution change, according to commonsense or domain knowledge. The settings of target shift [10, 14, 12, 15] and conditional shift [12, 16, 17] assume only $P(Y)$ and $P(X|Y)$ change, respectively, and their combination, as generalized target shift [12, 18], was also studied, and the corresponding methods clearly improved the performance on a number of benchmark datasets. The methods were extended further by learning feature representations with invariant conditionals given the label and matching joint distributions [17, 19, 20], and it was shown how methods based on domain-invariant representations can be understood from this perspective.

How are the distributions in different domains related? Essentially, DA aims to discover and exploit the constraints in the data distribution implied by multiple domains and make predictions that adapts to the target domain. To this end, we assume that the distributions of the data in different domains were independent and identically distributed (I.I.D.) drawn from some "mother" distribution. The mother distribution encodes the uncertainty in the domain-specific distributions, i.e., how the joint distribution is different across the domains. Suppose the mother distribution is known, from which the target-domain distribution is drawn. Furthermore, the target domain contains data points (without $Y$ values) generated by this distribution. It is then natural to leverage both the mother distribution and the target-domain feature values to reveal the property of the target-domain distribution for the purpose of predicting $Y$. In other words, DA is achieved by exploiting the mother distribution and the target-domain feature values to derive the information of $Y$.

Following this argument, we have several questions to answer. First, is there a natural, compact description of the constraints on the changes of the data distribution (to describe the mother distribution)? Such constraints include which factors of the joint distributions can change, whether they change independently, and the range of changes. (We represent the joint distribution as a product of the factors.) Second, how can we find such a description from the available data? Third, how can we make use of such a description as well as the target-domain data to make optimal prediction? Traditional graphical models have provided a compact way to encode conditional independence relations between variables and factorize the joint distribution [21, 22]. We will use an extension of Directed Acyclic Graphs (DAGs), called augmented DAGs, to factorize the joint distribution and encode which factors of the joint distribution change across domains. The augmented DAG, together with the conditional distributions and changeability of the changing modules, gives an augmented graphical model as a compact representation of how the joint distribution changes. Predicting the $Y$ values in the target domain is then a problem of Bayesian inference on this graphical model given the observed target-domain feature values. This provides a natural framework to address the DA problem in an automated, end-to-end manner.

## 2 Related Work

We are concerned with the scenario where no labeled point is available in the target domain, known as unsupervised DA. Various assumptions on how distribution changes were proposed to make successful knowledge transfer possible. For instance, a classical setting assumed that $P(X)$ changes but $P(Y|X)$ remains the same, i.e., the covariate shift situation; see, e.g., [1]. It is also known as sample selection bias in [2]. The correction of shift in $P(X)$ can be achieved by re-weighting source domain examples using importance weights as a function of feature $X$ [1, 2, 3, 4, 23, 24, 25, 26], based on certain distribution discrepancy measures such as Maximum Mean Discrepancy (MMD) [27]. A common prerequisite for such an approach is that the support for the source domain includes the target domain, but of course this is often not the case. Another collection of methods learns a domain-invariant representation by applying suitable linear transformation or nonlinear transformation or by properly sampling, which has identical distributions across the target and source domains [5, 6, 7, 8, 9].

In practice, it is very often that $P(X)$ and $P(Y|X)$ change simultaneously across domains. For instance, both of them are likely to change over time and location for a satellite image classification system. If the data distribution changes arbitrarily across domains, clearly knowledge from the sources may not help in predicting $Y$ in the target domain [28]. One has to find what type of

information should be transferred from sources to the target. A number of works are based on the factorization of the joint distribution as $P(XY) = P(X|Y)P(Y)$, in which either change in $P(Y)$ or in $P(X|Y)$ will cause changes in both $P(X)$ and $P(Y|X)$ according to the Bayes rule. One possibility is to assume the change in both $P(X)$ and $P(Y|X)$ is due to the change in $P(Y)$, while $P(X|Y)$ remains the same, as known as prior shift [10, 14] or target shift [12]. Similarly, one may assume conditional shift, in which $P(Y)$ remains the same but $P(X|Y)$ changes [12]. In practice, target shift and conditional shift may both happen, which is known as generalized target shift [12]. Various methods have been proposed to deal with these situations. Target shift can be corrected by re-weighting source domain examples using an importance function of $Y$, which can be estimated by density matching [12, 15, 29, 30, 31]. Conditional shift is in general ill-posed because without further constraints on it, $P^\tau(X|Y)$ is generally not identifiable given the source-domain data and the target-domain feature values. It has been shown to be identifiable when $P(X|Y)$ changes in some parametric ways, e.g., when $P(X|Y)$ changes under location-scale transformations of $X$ [12]. In addition, the invariant representation learning methods originally proposed for covariate shift can be adopted to achieve invariant causal mechanism [17]. Pseudo labels in the target domain may be exploited to refine the matching of conditional distributions [16, 32]. Finally, generalized target shift has also been addressed by joint learning of domain-invariant representations and instance weighting function; see e.g., [12, 17, 18, 33].

While the above works either assume $X \rightarrow Y$ or $Y \rightarrow X$, several recent works tried to model the complex causal relations between the features and label using causal graphs [34, 35, 36], e.g., a subset of features are the cause of $Y$ and the rest are effects. [34] presents a domain adaptation generative model according to the causal graph learned from data. [35] explores the features that have invariant conditional causal mechanisms for cross-domain prediction. [36] proposes an end-to-end method to transport invariant predictive distributions when a full causal DAG is unavailable. Our approach differs from these methods in two aspects. First, our method only requires the augmented DAG, which is easier to learn than a causal DAG. Second, our method can adapt both invariant and changing features, while [35] and [36] only exploit the features with invariant conditional distributions.

## 3 DA and Inference on Graphical Models

For the purpose of discover what to transfer in a automated way, in this paper we *mainly consider DA with at least two source domains*, although the method can be applied to the single-source case if proper additional constraints are known. Generally speaking, the availability of multiple source domains provides more hints helpful to find $P^\tau(X|Y)$ as well as $P^\tau(Y|X)$. Several algorithms have been proposed to combine source hypothesis from multiple source domains in different ways [37, 38, 39, 40]. As one may see, existing methods mainly assume the properties of the distribution shift and utilize the assumptions for DA; furthermore, the involved assumptions are usually rather strong. Violation of the assumptions may lead to negative transfer.

An essential question then naturally arises–is it possible to develop a data-driven approach to automatically figure out what information to transfer from the sources to the target and make optimal prediction in the target domain, under mild conditions? This paper aims at an attempt to answer this question, by representing the properties of distribution change with a graphical model, estimating the graphical model from data, and treating prediction in the target domain as a problem of inference on the graphical model given the target-domain feature values. Below we present the used graphical models and how to use them for DA.

### 3.1 Describing Distribution Change Properties with Augmented Graphical Models

In the target domain, the $Y$ values are to be predicted, and we aim at their optimal prediction with respect to the joint distribution. To find the target-domain distribution, one has to leverage source-domain data and exploit the connection between the distributions in different domains. It is then natural to factorize the joint distribution into different components or modules–it would facilitate recovering the target distribution if as few components as possible change across the domains. Furthermore, in estimation of the changing modules in the target domain, it will be beneficial if those changes are not coupled so that one can do "divide-and-conquer"; otherwise, if the changes are coupled, one has to estimate the changes together and would suffer from "curse-of-dimensionality". In other words, DA benefits from a compact description of how the data distribution can change across domains–such a description, together with the given feature values in the target domain, helps

recover the target joint distribution and enables optimal prediction. In this section we introduce our graphical model as such a way to describe distribution changes.

Traditional graphical models provide a compact, yet flexible, way to decompose the joint distribution of as a product of simpler, lower-dimensional factors [41, 22], as a consequence of conditional independence relations between the variables. For our purpose, we need encode not only conditional independence relations between the variables, but also whether the conditional distributions change across domains. To this end, we propose an augmented Directed Acyclic Graph (DAG) as a flexible yet compact way to describe how a joint distribution changes across domains, assuming that the distributions in all domains can be represented by such a graph. It is an augmented graph in the sense that it is over not only features $X_i$ and $Y$, but also external latent variables $\boldsymbol{\theta}$.

Figure 1 gives an example of such a graph. Nodes in gray are in the Markov Blanket (MB) of $Y$. The $\boldsymbol{\theta}$ variables are mutually independent, and take the same value across all data points within each domain and may take different values across domains. They indicate the property of distribution shift–for any variable with a $\boldsymbol{\theta}$ variable directly into it, its conditional distribution given its parents (implied by the DAG over $X_i$ and $\mathbf{Y}$) depends on the corresponding $\boldsymbol{\theta}$ variable, and hence may change across domains. In other words, the distributions across domains differ only in the values of the $\boldsymbol{\theta}$ variables. Once their values are given, the domain-specific joint distribution is given by $P(\mathbf{X}, Y \mid \boldsymbol{\theta})$, which can be factorized according to the augmented DAG. In the example given in Figure 1, distribution factors $P(X_1)$, $P(Y|X_1)$, and $P(X_3|Y, X_2)$, among others, change across domains, while $P(X_5|Y)$ and $P(X_7|X_3)$ are invariant. The joint data distribution in the $i$th domain can be written as

$$P(\mathbf{X}, Y|\boldsymbol{\theta}^{(i)}) = P(X_1|\theta_1^{(i)})P(Y|X_1, \theta_Y^{(i)})P(X_5|Y)P(X_2|Y, X_4, \theta_2^{(i)})P(X_3|Y, X_2, \theta_3^{(i)}) \times$$
$$P(X_4)P(X_6|X_4, \theta_6^{(i)})P(X_7|X_3).$$

We have several remarks to make on the used augmented graph. First, since the $\theta_i$ are independent, the corresponding conditional distributions change independently across domains. Because of such a independence property , one can model and learn the changes in the corresponding factors separately. Second, we note that each node in the augmented graph may be a set of variables, as a "supernode" instead of a single one. For instance, for the digit recognition problem, one can view the pixels of the digit image as such a "supernode" in the graph.

### 3.1.1 Relation to Causal Graphs

If the causal graph underlying the observed data is known, there is no confounder (hidden direct common cause of two variables), and the observed data are perfect random samples from the populations, then one can directly benefit from the causal model for transfer learning, as shown in [42, 12, 43]. In fact, in this case our graphical representation will encode the same conditional independence relations as the original causal model.

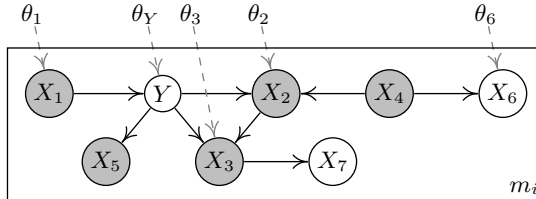

Figure 1: An augmented DAG over $Y$ and $X_i$. See main text for its interpretation.

It is worth noting that the causal model, on its own, might not be sufficient to explain the properties of the data, for instance, because of selection bias [44], which is often present in the sample. Furthermore, it is notoriously difficult to find causal relations based on observational data; to achieve it, one often has to make rather strong assumptions on the causal model (such as faithfulness [45]) and sampling process. On the other hand, it is rather easy to find the graphical model purely as a description of conditional independence relationships in the variables as well as the properties of changes in the distribution modules. The underlying causal structure may be very different from the augmented DAG we adopt. For instance, let $Y$ be disease and $X$ the corresponding symptoms. It is natural to have $Y$ as a cause of $X$. Suppose we have data collected in different clinics (domains) and that subjects are assigned to different clinics in a probabilistic way according to how severe the symptoms ($X$) are. Then one can see that across domains we have changing $P(X)$ but a fixed $P(Y|X)$ and, accordingly, in the augmented DAG has a directed link from $X$ to $Y$, contrary to the causal direction. For detailed examples as well as the involved causal graphs and augmented DAGs, please see Appendix A1.

## 3.2 Inference on Augmented Graphical Models for DA

We now aim to predict the value of $Y$ given the observed features $\mathbf{x}^\tau$ in the target domain, which is about $P(\mathbf{Y}^\tau \,|\, \mathbf{x}^\tau)$, where $\mathbf{Y}^\tau$ is the concatenation of $Y$ across all data points in the target domain. To this end, we have several issues to address. First, which features should be included in the prediction procedure? Second, as illustrated in Figure 1, a number of distribution factors change across domains, indicated by the links from the $\boldsymbol{\theta}$ variables, and it is not necessary to consider all of them for the purpose of DA–which changing factors should be adapted to the target-domain data?

Let us first show the general results on calculation of $P(\mathbf{Y}^\tau \,|\, \mathbf{x}^\tau)$, based on which prediction in the target domain is made. We then discuss how to simplify the estimator, thanks to the specific augmented graphical structure over $\mathbf{X}$ and $Y$. As the data are I.I.D. given the values of $\boldsymbol{\theta}$, we know $P(\mathbf{x}, \mathbf{y} \,|\, \boldsymbol{\theta}) = \prod_k P(\mathbf{x}_k, y_k \,|\, \boldsymbol{\theta})$ and $P(\mathbf{x} \,|\, \boldsymbol{\theta}) = \prod_k P(\mathbf{x}_k \,|\, \boldsymbol{\theta})$. Also bearing in mind that the value of $\boldsymbol{\theta}$ is shared within the same domain, we have

$$P(\mathbf{Y}^\tau = \mathbf{y}^\tau \,|\, \mathbf{x}^\tau) = \int P(\mathbf{y}^\tau \,|\, \mathbf{x}^\tau, \boldsymbol{\theta}) P(\boldsymbol{\theta}|\mathbf{x}^\tau) d\boldsymbol{\theta} \tag{1}$$

where $P(\boldsymbol{\theta}|\mathbf{x}^\tau) = \prod_k \left[ \sum_{y_k^\tau} P(\mathbf{x}_k^\tau, y_k^\tau \,|\, \boldsymbol{\theta}) \right] P(\boldsymbol{\theta}) / \int \prod_k \left[ \sum_{y_k^\tau} P(\mathbf{x}_k^\tau, y_k^\tau \,|\, \boldsymbol{\theta}) \right] P(\boldsymbol{\theta}) d\boldsymbol{\theta}$. For computational efficiency, we make prediction separately for different data points based on

$$P(y_k^\tau \,|\, \mathbf{x}^\tau) = \sum_{y_{k'}^\tau, \ k' \neq k} P(\mathbf{y}^\tau \,|\, \mathbf{x}^\tau) = \int P(y_k^\tau \,|\, \mathbf{x}_k^\tau, \boldsymbol{\theta}) P(\boldsymbol{\theta}|\mathbf{x}^\tau) d\boldsymbol{\theta}. \tag{2}$$

In the above expression, $P(\boldsymbol{\theta})$ is given in the augmented graphical model, $P(y_k^\tau, \mathbf{x}_k^\tau \,|\, \boldsymbol{\theta})$ can be calculated by using the chain rule on the augmented graphical model, Here we assume that the density $P(y_k^\tau, \mathbf{x}_k^\tau \,|\, \boldsymbol{\theta})$ is tractable, and we will show approximate inference procedures in Section 4.3 when we use implicit models to model $P(\mathbf{x}_k^\tau, y_k^\tau \,|\, \boldsymbol{\theta})$. Also, $P(y_k^\tau \,|\, \mathbf{x}_k^\tau, \boldsymbol{\theta})$ can be estimated by training a probabilistic classifier on the generated data from our model.

Moreover, for the purpose of predicting $Y$, not all $X_j$ are needed for the prediction of $Y$, and not all changing distribution modules need to adapt to the target domain. We exploit the graph structure to simplify the above expression. Let $\mathbf{V} = \mathbb{CH}(Y) \cup \{Y\}$, where $\mathbb{CH}(Y)$ denotes the set of children of $Y$ relative to the considered augmented DAG. Also denote by $\mathbb{PA}(V_j)$ the parent set of $V_j$. The conditional distribution of $V_j$ given its parents is $P(V_j \,|\, \mathbb{PA}(V_j), \theta_{V_j})$, where $\theta_{V_j}$ is the empty set if this conditional distribution does not change across domains. Let

$$\mathcal{C}_{jk} := P(v_{jk}^\tau \,|\, \mathbb{PA}(v_{jk}^\tau), \theta_{V_j}) \tag{3}$$

be shorthand for the conditional distribution of $V_j$ taking value $v_{jk}^\tau$ conditioning on its parents taking the $k$th value in the target domain and the value of $\theta_{V_j}$. $P(\mathbf{x}_k^\tau, y_k^\tau \,|\, \boldsymbol{\theta})$ can be factorized as

$$P(\mathbf{x}_k^\tau, y_k^\tau \,|\, \boldsymbol{\theta}) = \left[ \prod_{V_j \in \mathbf{V}} \mathcal{C}_{jk} \right] \cdot \underbrace{\left[ \prod_{W_j \notin \mathbf{V}} P(w_{jk}^\tau \,|\, \mathbb{PA}(w_{jk}^\tau), \theta_{W_j}) \right]}_{\triangleq N_k, \text{ which does not dependent on } y_k^\tau}.$$

Substituting the above expression into Eq. 2, one can see that $N_k$, defined above, will not appear in the final expression, so finally

$$P(y_k^\tau \,|\, \mathbf{x}^\tau) = \int P(y_k^\tau \,|\, \mathbf{x}_k^\tau, \boldsymbol{\theta}) \frac{\prod_k \left[ \sum_{y_k^\tau} \prod_{V_j \in \mathbf{V}} \mathcal{C}_{jk} \right] \prod_{V_j \in \mathbf{V}} P(\theta_{V_j})}{\int \prod_k \left[ \sum_{y_k^\tau} \prod_{V_j \in \mathbf{V}} \mathcal{C}_{jk} \right] \prod_{V_j \in \mathbf{V}} P(\theta_{V_j}) d\theta_{V_j}} d\boldsymbol{\theta}. \tag{4}$$

It is natural to see from the above final expression of $P(y_k^\tau \,|\, \mathbf{x}^\tau)$ that 1) only the conditional distributions for $Y$ and its children (variables in $\mathbf{V}$) need to be adapted (their corresponding $\boldsymbol{\theta}$ variables are involved in the expression) and that 2) among all features, only those in the MB of $Y$ are involved in the expression.

### 3.2.1 Benefits from a Bayesian Treatment

Many traditional procedures for unsupervised DA are concerned with the identifiability of the joint distribution in the unlabeled target domain [42, 12, 17]. If the joint distribution is identifiable, a

classifier can be learned by minimizing the loss with respect to the target-domain joint distribution. For instance, the so-called location-scale transformation is assumed for the features given the label $Y$ [12], rendering the target-domain joint distribution identifiable. Otherwise, successful DA is not guaranteed without further constraints. Even in the situation where the target-domain joint distribution is not identifiable, the Bayesian treatment, by incorporating the prior distribution of $\boldsymbol{\theta}$ and inferring the posterior of $Y$ in the target domain, may provide very informative prediction–the prior distribution of $\boldsymbol{\theta}$ constrains the changeability of the distribution modules, and such constraints may enable "soft" identifiability. For an illustrative example on this, please see Appendix A2.

## 4 Implementation of Data-Driven DA

In practice we are given data and the graphical model is often not available. For DA, we then need to learn (the relevant part of) the augmented graphical model from data, which includes the augmented DAG structure, the conditional distribution of each variable in $\mathbb{CH}(Y) \cup \{Y\}$ given its parents, and the prior distribution of the relevant $\boldsymbol{\theta}$ variables, and then develop computational methods for inferring $Y$ on it given the target-domain data.

### 4.1 Learning the Augmented DAG

For wide applicability of the proposed method, we aim to find a nonparametric method to learn the augmented DAG, instead of assuming restrictive conditional models such as linear ones. We note that in the causality community, finding causal relations from nonstationary or heterogeneous data has attracted some attention in recent years. In particular, under a set of assumptions, a nonparametric method to tackle this causal discovery problem, called Causal Discovery from NOn-stationary/heterogeneous Data (CD-NOD) [46, 47, 48], was recently proposed. The method is an extension of the PC algorithm [49] and consists of 1) figuring out where the causal mechanisms change, 2) estimation of the skeleton of the causal graph, and 3) determination of more causal directions compared to PC by using the independent change property of causal modules. Here we adapt their method for learning the portion of the augmented DAG needed for DA, without resorting to the assumptions made in their work.

Denote by $\mathbf{S}$ the set of $Y$ and all $X_i$. The adapted method has the following three steps. The first two are directly adapted from CD-NOD. *Step 1* is to find changing distribution factors and estimate undirected graph. Let $C$ be the domain index. It applies the first stage of the PC algorithm to $\mathbf{S} \cup \{C\}$ to find an undirected graph. It is interesting to note that if variable $S_i \in \mathbf{S}$ is adjacent to $C$, then $S_i$ is conditionally dependent on $C$ given *any* subset of the remaining variables, and hence, $P(S_i \mid \mathbb{PA}(S_i))$ must change across domains. Compared to the dataset shift detection method [50], our procedure is more general because it applies to multiple domains and can distinguish between invariant and changing conditional distribu-

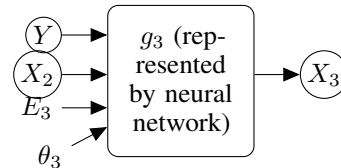

Figure 2: LV-CGAN for modeling $P(X_3 \mid Y, X_2, \theta_3)$ implied by the graph given in Figure 1.

tions and further even leverage useful information in the changing conditional distributions. *Step 2* is to determine edge directions, by applying the orientation rules in PC, with the additional constraints that all the $\boldsymbol{\theta}$ variables are exogenous and independent. Furthermore, if $S_i$ and $S_j$ are adjacent and are both adjacent to $C$, use the direction between them which gives independent changes in their conditional distributions, $P(S_i \mid \mathbb{PA}(S_i))$ and $P(S_j \mid \mathbb{PA}(S_j))$ [47]. If the changes are dependent in both directions, merge $S_i$ and $S_j$ as (part of) a "supernode'', and merge their corresponding $\boldsymbol{\theta}$ variables. *Step 3* finally Instantiates a DAG from the output of Step 2, which is a partially DAG. It is worth noting that our procedure is essentially local graph learning (focusing on only Y and variables in its Markov blanket). Thus, the complexity is not very sensitive to the original dimensions of the data, but to how large the Markov blanket is. For details of this procedure, see Appendix A3.

### 4.2 Latent-Variable CGAN for Modeling Changing Conditional Distributions

The second practical issue to be addressed is how to represent and learn the conditional distributions involved in (4). In some applications domain knowledge is available, and one may adopt specific models, like the Gaussian process model, that are expected to be suitable for the application domain. In this paper, in light of the power of the Generative Adversarial Network (GAN) [51] in capturing the property of high-dimensional distributions and generating new random samples and the capacity

of Conditional GAN (CGAN) [52] in learning flexible conditional distribution, we propose an extension of CGAN, namely, Latent-Variable CGAN (LV-CGAN), to model and learn a class of conditional distributions $P(S_i \,|\, \mathbb{PA}(S_i), \theta_{S_i})$, with $\theta_{S_i}$ as a latent variable. As an example, Figure 2 shows the structure of the LV-CGAN to model the conditional distribution of $P(X_3 \,|\, Y, X_2)$ across domains implied by the augmented DAG given in Figure 1. The whole network, including its parameters, is shared, and only the value of $\theta_3$ may vary across domains. Hence, it explicitly models both changing and invariant portions in the conditional distribution. In the $i$-th domain, $\theta_3$ takes value $\theta_3^{(i)}$ and encodes the domain-specific information. The network specifies a model distribution $Q(X_3|Y, X_2, \theta_3)$ by the generative process $X_3 = g_3(Y, X_2, E_3, \theta_3)$, which transforms random noise $E_3$ to $X_3$, conditioning on $Y$, $X_2$, and $\theta_3$. $E_3$ is independent of $Y$ and $X_2$, and its distribution is fixed (we used the standard Gaussian distribution). $g_3$ is a function represented by a neural network (NN) and shared by all domains. $Q(X_3|Y, X_2, \theta_3^{(i)})$ is trained to approximate the conditional distribution $P(X_3 \,|\, Y, X_2)$ in the $i$th domain. For invariant conditional distributions such as $P(X_5|Y)$ in Figure 1, the $\theta$ input vanishes and it becomes a CGAN. Compared to existing causal generative models [53, 54], which is learned on a single domain, our LV-CGAN aims to model the distribution changes across domains and generate labeled data in the target domain for cross-domain prediction.

### 4.3 Learning and Inference

Because we use GAN to model the distributions, the inference rules (2) and (4) are not be directly applied because of the intractability of the involved distributions. To tackle this problem, we develop a stochastic variational inference (SVI) [55] procedure to directly approximate the posterior $P(\theta|\mathbf{x}^\tau, \mathbf{y}^\tau)$ in the source domain and $P(\theta|\mathbf{x}^\tau)$ in the target domain. For simplicity of notation, we denote the $i$-th source domain data as $\mathcal{D}^i$, the target domain data as $\mathcal{D}^\tau$, and the combined source and target domain data as $\mathcal{D}$. We rely on the evidence lower bound (ELBO) of marginal likelihood in both source and target domains:

$$\log p(\mathcal{D}) \geq -\sum_{i=1}^{s} \mathrm{KL}(q(\theta|\mathcal{D}^i)|p(\theta)) + \mathbb{E}_{q(\theta|\mathcal{D}^i)}\Big[\sum_{k=1}^{m_i} \log p_g(\mathbf{x}_k^{(i)}, y_k^{(i)}|\theta)\Big]$$

$$-\mathrm{KL}(q(\theta|\mathcal{D}^\tau)|p(\theta)) + \mathbb{E}_{q(\theta|\mathcal{D}^\tau)}\Big[\sum_{k=1}^{m} \log p_g(\mathbf{x}_k^\tau|\theta)\Big]. \quad (5)$$

We approximate the posterior of $\theta$ in source and target domains with the Gaussian distribution $q(\theta|\mathcal{D}^i) = \mathcal{N}(\theta|\mu^{(i)}, \sigma^{(i)})$, $q(\theta|\mathcal{D}^\tau) = \mathcal{N}(\theta|\mu^\tau, \sigma^\tau)$. Then we can learn the model parameters in $g$ as well as the the variational parameters in each domain by the variational EM algorithm.

Up to now, we have followed the standard SVI procedure and assume that the density $p_g(X, Y, \theta)$ induced by the GAN generator $g$ is tractable, which is not true in our case. To extend the standard SVI for implicit distributions, we replace $\sum_{k=1}^{m_i} \log p_g(\mathbf{x}_k^{(i)}, y_k^{(i)}, \theta)$ with Jensen-Shannon divergence or Maximum Mean Discrepancy [56] that compares the empirical distributions of the $i$-th source domain data and the data generated from $g$. We perform the same procedure in the target domain. More details and theoretical justification of this procedure can be found in Appendix A4.

After learning the variational parameters, we can sample $\theta$ for the target domain and generate samples form $g$ to learn a classifier to approximate $P(y_k^\tau|\mathbf{x}^\tau)$. To make the procedure more efficient, we can make use of the decomposition of the joint distribution $p_g(X, Y, \theta)$ over the augmented graph, as shown in Eq. 4. The detailed derivations and justifications can be found in Appendix A5.

## 5 Experiments

### 5.1 Simulations

We simulate binary classification data from the graph on Figure 1, where we vary the number of source domains between 2, 4 and 9. We model each module in the graph with 1-hidden-layer MLPs with 32 nodes. In each replication, we randomly sample the MLP parameters and domain-specific $\theta$ values from $N(0, \mathbf{I})$. We sampled 500 points in each source domain and the target domain. We compare our approach, denoted by `Infer` against alternatives. We include a hypothesis combination method, denoted `simple_adapt` [37], linear mixture of source conditionals [13] denoted by `weigh` and `comb_classif` respectively. We also compare to the pooling SVM (denoted `poolSVM`), which merges all source data to train the SVM, as well as domain-invariant component analysis (DICA) [57], and Learning marginal predictors (LMP) [58]. The results are presented in Table 1. From the results,

Table 1: Accuracy on simulated datasets for the baselines and proposed method. The values presented are averages over 10 replicates for each experiment. Standard deviation is in parentheses.

| | DICA | weigh | simple_adapt | comb_classif | LMP | poolSVM | Infer |
|---|---|---|---|---|---|---|---|
| 9 sources | 80.04(15.5) | 72.1(14.5) | 70.0(14.3) | 72.34(16.24) | 78.90(13.81) | 71.8(11.43) | **83.90(9.02)** |
| 4 sources | 74.16(13.2) | 67.88(13.7) | 65.22(16.00) | 69.64(15.8) | 79.06(13.93) | 70.08(12.25) | **85.38(11.31)** |
| 2 sources | 86.56(13.63) | 75.04(18.8) | 69.42(17.87) | 74.28(18.2) | 84.52(13.72) | 83.84(13.7) | **93.10(7.17)** |

Table 2: Accuracy on the Wi-Fi & Flow data. Standard deviation is in parentheses.

| | DICA | weigh | LMP | poolSVM | Soft-Max | poolNN | Infer |
|---|---|---|---|---|---|---|---|
| t2,t3 → t1 | 29.32(2.5) | 43.71(3.02) | 46.80(1.4) | 40.25(1.6) | 44.86(5.1) | 42.88(1.6) | **70.8(2.7)** |
| t1,t3 → t2 | 24.5(3.6) | 38.19(1.9) | 39.11(2.1) | 48.70(1.8) | 44.95(4.4) | 47.41(2.1) | **84.5(2.9)** |
| t1,t2 → t3 | 21.7(3.9) | 36.03(1.85) | 39.28(2.05) | 40.46(1.4) | 43.63(4.1) | 41.00(1.8) | **83.0(7.3)** |
| Flow 3 sources | 79.2(11.0) | 84.2(9.3) | 91.6 (8.4) | 92.1(7.5) | 89.0(9.7) | 95.7(5.2) | **96.8(3.5)** |
| Flow 5 sources | 83.1(12.0) | 92.9(7.0) | 92.3 (6.4) | 94.7(6.1) | 89.7(8.0) | 96.0(5.1) | **97.1(3.5)** |

it can be seen that the proposed method outperforms the baselines by a large margin. Regarding significance of the results, we compared our method with the two other most powerful methods (DICA and LMP) using Wilcoxon signed rank test. The the p-values are 0.074, 0.009, 0.203 (against DICA) and 0.067, 0.074, 0.074 (against LMP), for 2, 4, and 9 source domains, respectively.

## 5.2 Wi-Fi Localization Dataset

We then perform evaluations on the cross-domain indoor WiFi location dataset [59]. The WiFi data were collected from a building hallway area, which was discretized into a space of grids. At each grid point, the strength of WiFi signals received from $D$ access points was collected. We aim to predict the location of the device from the $D$-dimensional WiFi signals. For the multiple-source setting, we cast it as a classification problem, where each location is assigned with a discrete label. We consider the task of transfer between different time periods, because the distribution of signal strength changes with time while the underlying graphical model is rather stable, which satisfies our assumption. The WiFi data were collected by the same device during three different time periods t1, t2, and t3 in the same hallway. Three sub-tasks including t2, t3 → t1, t1, t3 → t2, and t1, t2 → t3 are taken for performance evaluation. We thus obtained 19 possible labels, and in each domain we sampled 700 points in 10 replicates. We learn the graphical model and changing modules from the two source domains, and perform learning and Bayesian inference in all the domains. It took around six hours on the Wifi data with 69 variables. The graph learned from the Wifi t1 and t2 data is given in the Appendix A6. We implement our LV-CGAN by using Multi-Layer Perceptions (MLPs) with one hidden layer (32 nodes) to model the function of each module and set the dimension of input noise $E$ and $\theta$ involved in each module to 1. The reported result is classification accuracy of location labels. We use the same baselines as in the simulated dataset, excluding simple_adapt and comb_classif, and add a stronger baseline poolNN which replaces SVM in poolSVM with NN. We also compare with a recent adversarial learning method Soft-Max [60]. We present the results in Table 2. The results show that our method outperforms all baselines by a large margin.

## 5.3 Flow Cytometry Dataset

We also evaluate our method on the Graft vs. Host Disease Flow Cytommetry dataset (GvHD) [61]. The dataset consists of blood cells from patients, and the task is to classify each cell whether it is a lymphocite based on cell surface biomarkers. It is reasonable to assume that each patient has a different distribution of cells, and being able to predict the cell type in a new unlabeled patient given existing labeled patient data is an important task. There are 29 patients with 7 cell surface biomarkers, and we performed 29 experiments for each patient, where we treat it as a target domain subsample rest of the patients as source domains. We use the same baseline methods as in the Wifi dataset. We present classification accuracy results for 3 and 5 source domains in Table 2. The results show that our method is much better than most of the methods and performs slightly better than poolNN, which is a very strong baseline on this dataset.

## 5.4 Digits Datasets

Following the experimental setting in [60], we build a multi-source domain dataset by combing four digits datasets, including MNIST, MNIST-M, SVHN, and SynthDigits. We take MNIST, MNIST-M, and SVHN in turn as the target domain and use the rest domains as source domains, which leads to three domain adaptation tasks. We randomly sample 20,000 labeled images for training in the source domain, and test on 9,000 examples in the target domain. We use $Y \rightarrow X$ (as in previous work such

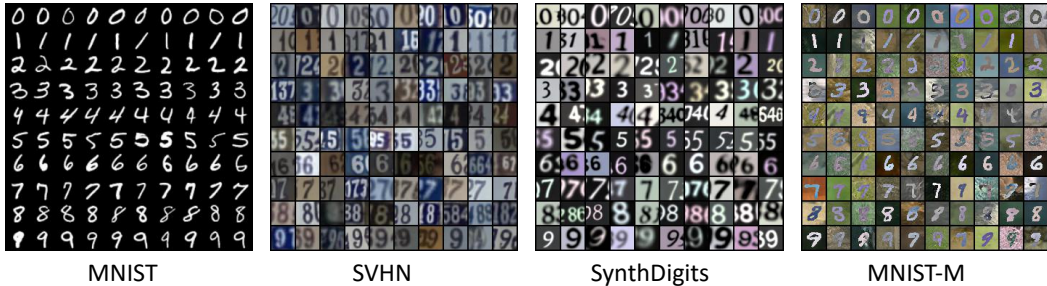

| MNIST | SVHN | SynthDigits | MNIST-M |
|---|---|---|---|

Figure 3: The generated images in each domain in the T+S+D/M task. Each row of an image corresponds to a fixed $Y$ value, ranging from 0 to 9. MNIST-M is the unlabeled target domain while the rest are source domains.

as [17]), where $X$ is the image, as the graph for adaptation. We leverage a recently proposed twin auxiliary classifier GAN framework [62] to match conditional distributions of generated and real data. More implementation details can be found in the Appendix A7.

We compare our method with recent deep multi-source adaptation method MDAN [60], with two variants Hard-Max and Soft-Max, and several baseline methods evaluated in [60], including `poolNN` and denoted `weight` described above and `poolDANN`) that considers the combined source domains as a single source domain and perform the DANN method [9]. Because our classifier network is different from that used in [60], we also report the `poolNN` method with our network architecture, denoted as `poolNN_Ours`.

The quantitative results are shown in Table 3. It can be seen that our method achieves much better performance than alternatives on the two hard tasks. This is very impressive because our baseline classifier (`poolNN_Ours`) performs worse `poolNN` in [60]. Figure 3 shows the generated images in each domain in the T+S+D/M task. Each row of an image corresponds to a fixed $Y$ value, ranging from 0 to 9. It can be seen that our method generates correct images for the corresponding labels, indicating that our method successfully transfer label knowledge from source domains and recovers the conditional distribution $P(X|Y)$ (also $P(Y|X)$) in the unlabeled target domain. The generated images for the other two tasks are given in the Appendix A8.

Table 3: Accuracy on the digits data. T: MNIST; M: MNIST-M; S: SVHN; D: SynthDigits.

|  | weigh | poolNN | poolDANN | Hard-Max | Soft-Max | poolNN_Ours | Infer |
|---|---|---|---|---|---|---|---|
| $S+M+D/T$ | 75.5 | 93.8 | 92.5 | 97.6 | **97.9** | 94.9 | 96.64 |
| $T+S+D/M$ | 56.3 | 56.1 | 65.1 | 66.3 | 68.7 | 59.6 | **89.89** |
| $M+T+D/S$ | 60.4 | 77.1 | 77.6 | 80.2 | 81.6 | 67.8 | **89.34** |

## 6 Conclusion and Discussions

In this paper, we proposed a framework to deal with unsupervised domain adaptation with multiple source domains by considering domain adaptation as an inference problem on a particular type of graphical model over the target variable and features or their combinations as super-nodes, which encodes the change properties of the data across domains. The graphical model can be directly estimated from data, leading to an automated, end-to-end approach to domain adaptation. As future work, we will study how the sparsity level of the learned graph affects the final prediction performance and, more importantly, aim to improve the computational efficiency of the method by resorting to more efficient inference procedures. Dealing with transfer learning with different feature spaces (known as heterogeneous transfer learning) by extending our approach is also a direction to explore.

## Acknowledgement

We are grateful to the anonymous reviewers, whose comments helped improve the paper. KZ would like to acknowledge the support by the United States Air Force under Contract No. FA8650-17-C-7715.

## Broader Impact

Domain adaptation aims to learn predictive models that can generalize to new domains that have different distribution than the training distributions. It is an essential step towards more generalizable and adaptive learning paradigms. We propose a brand new domain adaptation framework based on the graphical model that encodes conditional independence as well as distribution change properties. Our framework will inspire more effective DA algorithms that take advantage of the underlying data generating process. Open source algorithms and codes will benefit science, society, and the economy internationally through the further applications to analyzing social, business, and health data.

The research may greatly benefit practitioners in industry communities, where large amounts of unlabeled and heterogeneous data are ubiquitous. It will greatly save the expenses to label new datasets once some characteristics of the data changes. For example, a disease diagnosis model can be easily adapted to new hospitals without much labeling effort. A possible negative effect is that data annotators may lose their job.

The proposed method does not leverage any bias in the data.

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
