[Supplementary Material]

# Appendix of "Domain Adaptation as a Problem of Inference on Graphical Models"

## A1. Examples to Illustrate the Difference between Causal Graph and Our Augmented DAG

(a) The underlying data generating process of Example 1. $Y$ generates (causes) $X$, and $S$ denotes the selection variable (a data point is included if and only if $S = 1$).

(b) The augmented DAG representation for Example 1 to explain how the data distribution changes across domains.

(c) The generating process of Example 2. $L$ is a confounder; the mechanism of $X$ changes across domains, as indicated by $\eta_X$.

(d) The augmented DAG representation for Example 2 to explain how the data distribution changes across domains.

Figure 1: Two examples to illustrate the difference between the underlying causal graph and the augmented DAG used to represent the property of distribution changes across domains. (a) and (c) are the causal graphs of the two examples, and (b) and (d) the corresponding augmented DAGs.

Here we give two simple examples to illustrate the possible difference between the underlying causal structure and the graph we use for domain adaption. In Example 1, let $Y$ be disease and $X$ the corresponding symptoms. It is natural to have $Y$ as a cause of $X$. Suppose we have data collected in difference clinics, each of which corresponds to a domain. Further assume that subjects are assigned to different clinics in a probabilistic way according to how severe the symptoms are. Figure 1(a) gives the causal structure together with the sampling process to generate the data in each domain. $S$ is a selection variable, and a data point is selected if and only $S$ takes value 1. $P(S = 1|X)$ depends on $\eta_S$, which may take different values across domains, reflecting different sampling mechanisms (e.g., subjects go to different clinics according to their symptoms). In this case, according to data in different domains, $P(X)$ changes. But $P(Y|X)$ will stay the same because according to the process given in (a), $Y$ and $S$ are conditionally independent given $X$ and, as a consequence, $P(Y|X, S) = P(Y|X)$. The graphical model for describing the distribution change across domains is given in 1(b)–they are apparently inconsistent, and the direction between $Y$ and $X$ is reversed; however, for the purpose of DA, the graph in (b) suffices and, furthermore, as shown later, it can be directly learned from data from multiple domains. Example 2 follows the causal structure given in Figure 1(c), where $X$ and $Y$ are not directly causally related but have a hidden direct common cause (confounder) $L$ and the generating process of $X$ also depends on $\eta_X$, whose value may vary across domains. We care only about how the distribution changes–since in this example $P(Y)$ remains the same across domains, we can factorize the joint distribution as $P(Y, X) = P(Y)P(X|Y)$, in which only $P(X|Y)$ changes across domains, and the corresponding augmented DAG is shown in (d).

(a) Prior distributions of $\boldsymbol{\theta}$        (b) Posterior of $\theta_Y$ given $\mathbb{V}\mathrm{ar}(X)$

Figure 2: An illustration of the benefit of Bayesian treatment of the changeability of distribution modules ( represented by the $\boldsymbol{\theta}$ variables).

## A2. Illustration of Benefits from a Bayesian Treatment

Here is an example showing the benefits of a Bayesian treatment. For clarity purposes, we use simple parametric models and a single feature $X$ for the conditional distributions: $Y \sim \mathcal{N}(0, \theta_Y)$, $X = Y + E$, where $E \sim \mathcal{N}(0, \theta_X)$, i.e., $X|Y \sim \mathcal{N}(Y, \theta_2)$. So $\theta_Y$ controls the distribution of $Y$, and $\theta_X$ controls the conditional distribution of $X$ given $Y$. The marginal distribution of $X$ is then $X \sim \mathcal{N}(0, \theta_Y + \theta_X)$, which is what we can observe in the target domain. Clearly, from $P(X)$ in the target domain, $P(Y)$ or $P(X|Y)$ is not identifiable because $P(X)$ gives only $\theta_Y + \theta_X$. Now suppose we have prior distributions for $\theta_Y$ and $\theta_X$: $\theta_Y \sim \Gamma(3, 1)$ and $\theta_X \sim \Gamma(1.5, 1)$, where the two arguments are the shape and scale parameters of the gamma distribution, respectively. Figure 2(a) shows their prior distributions, and (b) gives the corresponding posterior distribution of $\theta_Y$ given the variance of $X$, whose empirical version is observed in the target domain. One can see that although $\theta_Y$ as well as $\theta_X$ is not theoretically identifiable, $P(\theta_Y \,|\, \mathbb{V}\mathrm{ar}(X))$ is informative as to the value that $\theta_Y$ may take. Especially when $\mathbb{V}\mathrm{ar}(X)$ is relatively small, the posterior distribution is narrow. The information we have about $\theta_Y$ and $X$ then allows non-trivial prediction of the target-domain joint distribution and the $Y$ values from the values of $X$.

## A3. The Procedure of Learning the Augmented DAG

Denote by $\mathbf{S}$ the set of $Y$ and all $X_i$. The adapted DAG learning method has the following three steps.

**Step 1 (Finding changing distribution factors and estimating undirected graph)**   Let $C$ be the domain index. Apply the first stage of the PC algorithm to $\mathbf{S} \cup \{C\}$ (the domain index $C$ is added to the variable set to capture the changeability of the conditional distributions); it starts with an undirected, fully connected graph, removes the edge between two variables that are conditionally independent given some other variables, and finally determines the skeleton. It is interesting to note that if variable $S_i \in \mathbf{S}$ is adjacent to $C$, then $S_i$ is conditionally dependent on $C$ given *any* subset of the remaining variables, and hence, there exists two different values of $C$, $c_1$ and $c_2$, such that $P(S_i \,|\, \mathbb{PA}(S_i), C = c_1) \neq P(S_i \,|\, \mathbb{PA}(S_i), C = c_2)$, meaning that $P(S_i \,|\, \mathbb{PA}(S_i))$ must change across domains. Also add variable $\theta_{S_i}$ in the graph, which points to $S_i$.

**Step 2 (Determining edge direction with additional constraints)**   We then find v-structures in the graph and do orientation propagation, as in the PC algorithm [1], but we benefit from additional constraints implied by the augmented DAG structure. In this procedure, we first make use of the constraint that if variable $S_i$ is adjacent to $C$, then there exists a $\theta$ variable, $\theta_{S_i}$, pointing to $S_i$; given this direction, one may further determine the directions of other edges [2]. In particular, suppose $S_j$ is adjacent to $S_i$ but not to $C$. Then if it is conditionally independent from $C$ given a variable set that does not include $S_i$, orient the edge between them as $S_j \to S_i$; if it is conditionally

independent from $C$ given a variable set that includes $S_i$, orient it as $S_j \leftarrow S_i$. Second, if $S_i$ and $S_j$ are adjacent and are both adjacent to $C$, use the direction between them which gives independent changes in their conditional distributions, $P(S_i \mid \mathbb{PA}(S_i))$ and $P(S_j \mid \mathbb{PA}(S_j))$ [3]. If the changes are dependent in both directions, merge $S_i$ and $S_j$ as (part of) a "supernode" in the graph, and merge their corresponding $\boldsymbol{\theta}$ variables.

**Step 3 (Instantiating a DAG)** Step 2 produces a partially directed acyclic graph (PDAG), representing an Markov equivalence class [4]. All augmented DAGs in this equivalence class have the same (conditional) independence relations, so finally, we instantiate from the equivalence class a DAG over $Y$ and the variables in its Markov Blanket (MB). (It was shown in Section 2 that inferring the posterior of $Y$ involves only the conditional distributions of $Y$ and its children, not necessarily the conditional distribution of every feature.)

Two remarks are worth making on this procedure. First, to avoid strong assumptions on the forms of the conditional distributions, we make use of a nonparametric test of conditional independence, namely, kernel-based conditional independence test [5], when learning the augmented DAG. Second, given that the final inference for $Y$ in the target domain depends only on the conditional distributions of $Y$ and its children, one may extend some form of local graph structure discovery procedure (see, e.g., [6, 7]), to directly find the local graph structure involving $Y$ and variables in its MB. This will be particularly beneficial on the computational load if we deal with high-dimensional features.

## A4. Stochastic Variational Inference for Latent-Variable Conditional GAN

For better illustration of the inference procedure, we consider the situation where we do not use the graphical relations between $X_i$ and $Y$. In this case, the data in all domains can be modeled by a specific LV-CGAN $(X, Y) = g(E, \boldsymbol{\theta})$ with no condition variables. Once we have knowledge about the graphical model, either from domain prior or by learning, we can breakdown the generator into a series of LV-CGANs according to the graph. The details will be given in the next section. For now, we consider the learning and inference in a general generative model.

The log-likelihood terms in Eq. (5) can be considered as empirical estimation of the Kullback–Leibler (KL) divergence between the data distribution and model distribution. Specifically, the KL divergence between joint distribution of $X$ and $Y$ in the $i$th source domain and the model distribution $p_g(X, Y | \boldsymbol{\theta})$ implied by the GAN generator $g$ (with $\boldsymbol{\theta}$ as an input) can be calculated by

$$
\begin{aligned}
&\texttt{KL}(P^{(i)}(X, Y) || p_g(X, Y | \boldsymbol{\theta})) \\
&= \int P^{(i)}(\mathbf{x}, y) \log P^{(i)}(\mathbf{x}, y) d\mathbf{x} dy - \int P^{(i)}(\mathbf{x}, y) \log p_g(\mathbf{x}, y | \boldsymbol{\theta}) d\mathbf{x} dy \\
&= c_i - \int P^{(i)}(\mathbf{x}, y) \log p_g(\mathbf{x}, y | \boldsymbol{\theta}) d\mathbf{x} dy,
\end{aligned}
\tag{A1}
$$

where the term $c_i$ is considered as a constant because it does not contain any model parameters in $g$. The empirical estimation of $\texttt{KL}(P^{(i)}(X, Y) || p_g(X, Y | \boldsymbol{\theta}))$ is

$$
\widehat{\texttt{KL}}(P^{(i)}(X, Y) || p_g(X, Y | \boldsymbol{\theta})) = \hat{c}_i - \frac{1}{m_i} \sum_{k=1}^{m_i} \log p_g(\mathbf{x}_k^{(i)}, y_k^{(i)} | \boldsymbol{\theta}),
\tag{A2}
$$

where $\hat{c}_i$ is an empirical estimation of $c_i$. Similarly, we have the KL divergence between the marginal distribution of $X$ in the target domain and the marginal distribution of $X$ induced by the GAN generator $g$:

$$
\widehat{\texttt{KL}}(P^\tau(X) || p_g(X | \boldsymbol{\theta})) = \hat{c}_\tau - \frac{1}{m} \sum_{k=1}^{m} \log p_g(\mathbf{x}_k^\tau | \boldsymbol{\theta}).
\tag{A3}
$$

For simplicity of notations, we assume all the source domains are of the same sample size, *i.e.*, $m_1 = m_2 = \ldots = m_s = m$. If the sample sizes of the domains are different, we can apply biased batch sampling, which samples the same number of data points from each domain in a mini-batch.

By multiplying both sides of Eq (5) by $\frac{1}{m}$ and adding the constants $-\hat{c}_i$ and $-\hat{c}_\tau$, we have

$$\frac{1}{m}\log p(\mathcal{D}) - \sum_{i=1}^{s}\hat{c}_i - \hat{c}_\tau \geq -\frac{1}{m}\sum_{i=1}^{s}\text{KL}(q(\boldsymbol{\theta}|\mathcal{D}^i)|p(\boldsymbol{\theta})) - \mathbb{E}_{q(\boldsymbol{\theta}|\mathcal{D}^i)}\left[\widehat{\text{KL}}(P^{(i)}(X,Y)||p_g(X,Y|\boldsymbol{\theta}))\right]$$

$$-\frac{1}{m}\text{KL}(q(\boldsymbol{\theta}|\mathcal{D}^\tau)|p(\boldsymbol{\theta})) - \mathbb{E}_{q(\boldsymbol{\theta}|\mathcal{D}^\tau)}\left[\widehat{\text{KL}}(P^\tau(X)||p_g(X|\boldsymbol{\theta}))\right]. \quad (A4)$$

Since $p_g(X,Y|\boldsymbol{\theta})$ and $p_g(X|\boldsymbol{\theta})$ are implied by a GAN generator $g$, we cannot compute the $\widehat{\text{KL}}$ terms in Eq. (A4). Instead, we replace the KL divergence with Maximum Mean Discrepancy (MMD) or Jensen-Shannon Divergence (JSD) that can compare the distributions of real data and the fake data generated from $g$. Specifically, given data $(\mathbf{x}_k^{(i)}, y_k^{(i)})_{k=1}^{B}$ from the $i$th source domain, and data $(\hat{\mathbf{x}}_k^{(i)}, \hat{y}_k^{(i)})_{k=1}^{B}$ from $g(\cdot, \boldsymbol{\theta})$ (where $B$ is the batch size), we have the following objective:

$$\max_{g,q} -\frac{1}{m}\sum_{i=1}^{s}\text{KL}(q(\boldsymbol{\theta}|\mathcal{D}^i)|p(\boldsymbol{\theta})) - \mathbb{E}_{q(\boldsymbol{\theta}|\mathcal{D}^i)}\left[\widehat{\text{Div}}(P^{(i)}(X,Y)||p_g(X,Y|\boldsymbol{\theta}))\right]$$

$$-\frac{1}{m}\text{KL}(q(\boldsymbol{\theta}|\mathcal{D}^\tau)|p(\boldsymbol{\theta})) - \mathbb{E}_{q(\boldsymbol{\theta}|\mathcal{D}^\tau)}\left[\widehat{\text{Div}}(P^\tau(X)||p_g(X|\boldsymbol{\theta}))\right]., \quad (A5)$$

where $\text{Div}$ can be MMD, JSD or any other divergence measures that can measure the distance between the real and fake samples. The empirical MMD between real and fake data is defined as

$$\widehat{\text{MMD}}(P^{(i)}(X,Y)||p_g(X,Y|\boldsymbol{\theta})) = \frac{1}{B^2}\sum_{k=1}^{B}\sum_{k'=1}^{B}k(\mathbf{x}_k^{(i)}, \mathbf{x}_{k'}^{(i)})l(y_k^{(i)}, y_{k'}^{(i)}) -$$

$$\frac{2}{B^2}\sum_{k=1}^{B}\sum_{k'=1}^{B}k(\mathbf{x}_k^{(i)}, \hat{\mathbf{x}}_{k'}^{(i)})l(y_k^{(i)}, \hat{y}_{k'}^{(i)}) + \frac{1}{B^2}\sum_{k=1}^{B}\sum_{k'=1}^{B}k(\hat{\mathbf{x}}_k^{(i)}, \hat{\mathbf{x}}_{k'}^{(i)})l(\hat{y}_k^{(i)}, \hat{y}_{k'}^{(i)}),$$

where $k$ and $l$ are kernel functions for $\mathbf{x}$ and $y$, respectively. The target-domain empirical MMD is of the same form except that $l$ function needs to be removed because only marginal distributions of $X$ are compared. According to the GAN formulation [8, 9], Jensen-Shannon Divergence (JSD) can be implemented by introducing a discriminator $D$:

$$\widehat{\text{JSD}}(P^{(i)}(X,Y)||p_g(X,Y|\boldsymbol{\theta})) = \max_D \frac{1}{B}\sum_{k=1}^{B}[\log D(\mathbf{x}_k, y_k)] + \frac{1}{B}\sum_{k=1}^{B}[\log(1 - D(\hat{\mathbf{x}}_k, \hat{y}_k))].$$

The target-domain JSD can be obtained by omitting $y_k$ and $\hat{y}_k$ in the formulation.

Finally, we can make use of Eq. A5 to learn the posterior distribution $q$ and generator $g$ in an end-to-end manner. For the expectation $\mathbb{E}_{q(\boldsymbol{\theta}|\mathcal{D}^i)}[\cdot]$, we use the reparameterization trick [10] and sample $\theta_j^{(i)}$ from the model $\boldsymbol{\theta} = \mu^{(i)} + \epsilon * \sigma^{(i)}$, where $\epsilon$ is a standard normal variable, such that the variational parameters in the posterior distribution of $\boldsymbol{\theta}$ can be simultaneously learned with $g$. If assuming the prior $p(\boldsymbol{\theta}) = \mathcal{N}(0, \mathbf{I})$, the KL divergence terms $\text{KL}(q(\boldsymbol{\theta}|\mathcal{D}^{(i)})|p(\boldsymbol{\theta}))$ have the following closed form solution:

$$\text{KL}(q(\boldsymbol{\theta}|\mathcal{D}^{(i)})|p(\boldsymbol{\theta})) = \frac{1}{2}\sum_{j=1}^{d}(-1 - \log((\sigma_j^{(i)})^2) + (\mu_j^{(i)})^2 + (\sigma_j^{(i)})^2), \quad (A6)$$

where $d$ is the dimensionality of $\boldsymbol{\theta}$. $\text{KL}(q(\boldsymbol{\theta}|\mathcal{D}^\tau)|p(\boldsymbol{\theta}))$ can be calculated in the same way.

## A5. Factorized Inference and Learning According to the Augmented DAG

In the previous section, we have shown the approximate inference and learning procedure when not taking into consideration of graph structure. In this section, we will show how the inference and learning procedure can be simplified given an augmented DAG. According to Eq. 4, we only need to consider the set $\mathbf{V} = \mathbb{CH}(Y) \cup \{Y\}$ for prediction of $Y$ in the target domain. According to an augmented DAG, we have the following factorization $P(\mathbf{V}|\boldsymbol{\theta}) = \prod_{V_j \in \mathbf{V}} P(V_j | \mathbb{PA}(V_j), \theta_{V_j})$.

According to the factorization, we can calculate the posterior of $\boldsymbol{\theta}$ in the $i$th source domain as

$$P(\boldsymbol{\theta}|\mathcal{D}^i) = \frac{\prod_k \prod_{V_j \in \mathbf{V}} P(v_{jk}^{(i)} \,|\, \mathbb{PA}(v_{jk}^{(i)}), \theta_{V_j}) P(\theta_{V_j})}{\int \prod_k \prod_{V_j \in \mathbf{V}} P(v_{jk}^{(i)} \,|\, \mathbb{PA}(v_{jk}^{(i)}), \theta_{V_j}) P(\theta_{V_j}) d\theta_{V_j}}$$

$$= \frac{\prod_{V_j \in \mathbf{V}} \prod_k P(v_{jk}^{(i)} \,|\, \mathbb{PA}(v_{jk}^{(i)}), \theta_{V_j}) P(\theta_{V_j})}{\prod_{V_j \in \mathbf{V}} \prod_k \int P(v_{jk}^{(i)} \,|\, \mathbb{PA}(v_{jk}^{(i)}), \theta_{V_j}) P(\theta_{V_j}) d\theta_{V_j}}$$

$$= \prod_{V_j \in \mathbf{V}} P(\theta_{V_j}|\mathcal{D}^i). \tag{A7}$$

However, the target domain $\boldsymbol{\theta}$ posterior cannot be factorized in this way because the marginalization w.r.t. $y_k^\tau$, as shown in Eq. 4. Therefore, we can simplify the inference and learning procedure (A5) by making use of the factorization in the source domain as

$$\max_{g,q} - \frac{1}{m} \sum_{i=1}^{s} \sum_{j=1}^{|\mathbf{V}|} \mathtt{KL}(q(\theta_{V_j}|\mathcal{D}^i)|p(\theta_{V_j})) - \mathbb{E}_{q(\theta_{V_j}|\mathcal{D}^i)} \Big[ \widehat{\mathtt{Div}}(P^{(i)}(V_j, \mathbb{PA}(V_j)) || p_{g_j}(V_j, \mathbb{PA}(V_j)|\theta_{V_j})) \Big]$$

$$- \frac{1}{m} \mathtt{KL}(q(\boldsymbol{\theta}|\mathcal{D}^\tau)|p(\boldsymbol{\theta})) - \mathbb{E}_{q(\boldsymbol{\theta}|\mathcal{D}^\tau)} \Big[ \widehat{\mathtt{Div}}(P^\tau(X) || p_g(X|\boldsymbol{\theta})) \Big], \tag{A8}$$

where $p_{g_j}(V_j, \mathbb{PA}(V_j)|\theta_{V_j}))$ is the distribution specified by the LV-CGAN $V_j = g_j(E_j, \mathbb{PA}(V_j), \theta_{V_j})$ and $p_g(X|\boldsymbol{\theta})$ is the marginal distribution of $X$ specified by a composition of all the LV-CGANs $g_j$ according to the augmented DAG.

After obtaining the approximate posterior distribution $q(\boldsymbol{\theta}|\mathcal{D}^\tau)$ and the LV-CGAN generator, we can perform prediction in the target domain by approximating Eq. 4 as

$$P(y_k^\tau \,|\, \mathbf{x}^\tau) = \int P(y_k^\tau \,|\, \mathbf{x}_k^\tau, \boldsymbol{\theta}) q(\boldsymbol{\theta}|\mathcal{D}^\tau) d\boldsymbol{\theta}$$

$$\approx \frac{1}{L} \sum_{l=1}^{L} P(y_k^\tau \,|\, \mathbf{x}_k^\tau, \theta_l), \tag{A9}$$

where $\theta_l \sim q(\boldsymbol{\theta}|\mathcal{D}^\tau)$ and $P(y_k^\tau \,|\, \mathbf{x}_k^\tau, \theta_l)$ is estimated by training a softmax classifier on the data generated from the LV-CGAN generator with $\theta_l$ as inputs.

## A6. Learned Graph on WiFi Dataset

Figure 3 shows the learned augmented DAG on the WiFi localization dataset (t1 & t2). The graphs learned on t2 & t3 and t1 & t3 are almost identical to the graph shown in Figure 3.

## A7. Implementation Details in Digit Adaptation Experiments

To generate high-quality images, we build our model based on BigGAN [11]. We choose a simple architecture and the generator and discriminator used are shown in Table 1.

For convenience, we use the following abbreviation: C = Feature channel, K = Kernel size, S = Stride size, SNLinear = A linear layer with spectral normalization (SN), and SNResBlk = A residual block with SN. The dimensionality of input noise $E$ is 128.

Because the original CGAN formulation [9] that feeds the concatenation of the label and image into a discriminator $D$ usually cannot generate high-quality images, we utilize the recent Twin Auxiliary Classifier GAN (TAC-GAN) framework [12] to match conditional distributions of generated and real data in the source domains. Specifically, the formulation contains the following modules after the shared feature extraction residual blocks: 1) the discriminator SNLinear (D) to distinguish real and generated images, 2) the primary auxiliary classifier (AC) to predict class labels of real images in source domains, 3) the twin auxiliary classifier (TAC) to predict labels of all generated images from CGAN generator, 4) the primary domain classifier (DC) to predict domain labels of all data, and 5) the twin domain classifier (TDC) to predict domain labels of all generated images from the CGAN generator.

Figure 3: The augmented DAG learned on the WiFi `t1` and `t2` datasets. Pink nodes denote the changing modules and green ones denote the constant modules whose conditional distribution does not change across domains.

Table 1: Network architecture for digits adaptation.

| Generator | | | | |
|---|---|---|---|---|
| Index | Layer | C | K | S |
| 1 | SNLinear | 256*4 | | |
| 2 | Upsample SNResblk | 256 | 3 | 1 |
| 3 | Upsample SNResblk | 256 | 3 | 1 |
| 4 | Upsample SNResblk | 256 | 3 | 1 |
| 5 | Relu+SNConv+Tanh | 3 | 3 | 1 |
| Discriminator | | | | |
| 1 | Downsample SNResblk | 256 | 3 | 1 |
| 2 | Downsample SNResblk | 256 | 3 | 1 |
| 3 | Downsample SNResblk | 256 | 3 | 1 |
| 4 | AveragePooling | 256 | | |
| $5_1$ | SNLinear (D) | 1 | | |
| $5_2$ | SNLinear (AC) | 10 | | |
| $5_3$ | SNLinear (TAC) | 10 | | |
| $5_4$ | SNLinear (DC) | 4 | | |
| $5_5$ | SNLinear (TDC) | 4 | | |

## A8. The generated images in Digit Adaptation Experiments

Figure 4 and Figure 5 show the generated images in the S+M+D/T and M+T+D/S tasks, respectively. The last image is the generated image from LV-CGAN conditioned on the labels. Though the target domain is unlabeled, our method successfully transfers information from the labeled source domains and reconstruct the conditional distributions $P(X|Y)$ in the target domain.

Figure 4: The generated images in each domain in the S+M+D/T task. Each row of an image corresponds to a fixed $Y$ value, ranging from 0 to 9. MNIST is the unlabeled target domain while the rest are source domains.

Figure 5: The generated images in each domain in the M+T+D/S task. Each row of an image corresponds to a fixed $Y$ value, ranging from 0 to 9. SVHN is the unlabeled target domain while the rest are source domains.