[Reviews · NeurIPS 2020]

Review 1

Summary and Contributions: The authors propose to infer an augmented graphical model, augmented by latent variables adjacent to observed nodes that represent changes in the conditional distributions across domains. This structure is then used to infer latent variables on a target domain using available covariates to improve predictions in this unseen environment. From a computational perspective, the procedure starts with estimating the graph of conditional independences and latent variables with an existing algorithm and estimate conditional distributions inferred from the graph using GANs. Experiments seem to justify this approach, with better performance over competing algorithms across a variety of datasets.

Strengths: Domain adaptation must be pivotal for any algorithm looking to be applied in practice. The authors take a graphical model perspective to infer what information should be shared across domains, which is becoming increasingly popular and the connections with causality and generalization will be of interest to the broader community. Inferring latent variables that indicate changes between domains automatically, and that can be estimated from target data to find good estimators of conditional distributions is very interesting. In my opinion, the contribution seems novel even though not a lot of comparisons with related work are made in the paper (so my judgement may be wrong). The paper is well written and I appreciate the effort made to explain the intuition on the graphical model used in the algorithm, which differs from causality.

Weaknesses: My biggest concern is what kind of guarantees can be ensured in practice: the relationship between domains and the assumptions on test data are not explored. For instance is it fair to say that transfer can be expected to be successful if the target data distribution can be written as a convex combination of source distributions? What if changes in new data are broader, affecting nodes or mechanisms that do not vary among training domains? Thinking of this problem from a causal perspective would entail thinking of interventions, data generating mechanisms and assumptions on changes between domains. All of this is lacking even though causality is mentioned in the abstract and introduction. The theoretical grounding is minimal which is a downside because we have no guarantees (or even an intuition) that we will avoid negative transfer. I think this is a missed opportunity because causality gives us a formal language to talk about distribution shifts in terms of interventions which I was expecting in this paper. Finally the experiments are somewhat underwhelming, synthetic examples are tailored exactly to the assumptions made by the authors and it does not give a sense of how differences in training environments translate to performance differences. The image example is also a bit odd, with only two variables and an assumed causal direction between them. I'll give more details below.

Correctness: Yes the method seems correct.

Clarity: Yes.

Relation to Prior Work: Not enough contrast with related work in my opinion.

Reproducibility: Yes

Additional Feedback: 1) The authors claim "their method can be applied to the single-source case if proper additional constraints are known", I don't see how this may be the case and it is not further explored. 2) A second concern I have relates to the propoagation of errors from the graph learning approach, testing for conditional independence is a hard problem especially in the high-dimensional setting and understanding the influence of errors made in that stage would certainly make the argument stronger. Moreover, from a computational perspective I would venture to think that the proposed approach is expensive, can any comparisons be made to understand their practical applicability? 3) The performance results are counterintuitive. In Table 1, performance is best with 2 sources, when I would have thought that more sources would improve performance, and furthermore performance increases after it drops with the number of sources. Is there a reason for this behaviour? 4) I don't quite understand what it means to be learning a graphical model over images (It seems to me that only two variables are considered, X the image and Y the target digit). Also the flow cytometry example has a very limited number of variables, how would the method work if the data is high-dimensional (from a performance and computational time perspective)? Typos: - l72 discovering - l89 joint - l135 In fact. - l 145 different - l215 missing bracket - l399 directly ------------------------------------------------------------------------------------------- After reading the rebuttal I appreciate that the authors have tried to address some of my concerns. It remains unclear to me though what is meant by "mother" distribution or even how it plays any role in the development of the method. I think this is important because the success of transfer learning invariably depends on the relationship between available domains, without understanding better the data generating process and what kind of deviations at test time are allowed its hard to judge the potential applications of this method in practice. Synthetic examples analysing potential scenarios would have been helpful and I hope that further iterations of this work will include wider empirical evaluations. For this reasin, at this time, I stand by my score.


Review 2

Summary and Contributions: The paper proposes a novel end-to-end methodology for unsupervised domain adaptation with multiple sources (even if the case of one single source is discussed). The idea of the paper is to describe the various domains by a single graphical model the hyperparameters of which depend on the domain. The overall method relies on (1) causal discovery method adapted from the recent CD-NOD algorithm for estimation of the graph; (2) GAN and CGAN for estimation of the distributions in the graphical model; (3) stochastic variational inference for inference. The method offers good results on both artificial and real data.

Strengths: The paper presents a very interesting and novel (to my knowledge) end-to-end Bayesian methodology for multi-source unsupervised domain adaptation. This methodology is rather (see my questions below) general and technically sound. The paper shows a good understanding of the problem of unsupervised domain adaptation and proposes interesting discussions on various points which are of direct interest for a full understanding of the presented method. The presented Bayesian model is extremely sensible and adapted to the tackled problem, even if I would have expected a more detailed discussion of the validity of the hypotheses of the model.

Weaknesses: As exposed in the strengths, I would have expected some more extensive discussion on the limitations of the presented method. The overall paper relies on the assumptions that (1) the distribution p(X,Y) can be factorized as a graphical model; (2) the graphical models have the same structure in all tasks. The second assumption seems very arguable in real cases. What is the level of generality of this assumption? And what does happen when it is not satisfied? Two cases can be observed: when one (or more) of the source tasks has a different (but close) structure, and when the target task has a different structure. These questions are very important in the theoretical literature of domain adaptation, where a major question is to relate the performance to a discrepancy between task distributions. I understand that theoretical results might be difficult to derive in the context of this research, but I would be interested in having an empirical evidence of the influence of discrepancy onto the performance. This could be done for instance with the simulated dataset of Section 4.1, where it would be easy to use a different graphical model for the target dataset. It would be interesting to observe how robust your methodology is toward changes in the target distribution. Another weakness of the method is the complexity of the method which is never fully discussed. The scalability to datasets with a large number of features seems quite difficult, in particular because of the learning of the augmented DAG. The CGAN step also must be quite costly. Could you please comment on this point, and also add the computation times in the experimental section? Also, the proposed method reconstructs the graphical models for the different sources. I would have expected to read a couple of words on how the graphs were inferred for instance in the simulated dataset. Is the model actually retrieved? For the real datasets, it could be used to visualize the relevant features (here the parents of y in the graph). Could you please comment on this point? Did you run any experiment to observe this? If so, could you provide some examples of the obtained results, for instance in the MNIST task. I wonder what the graphs are in this case. Last remark: I would have expected in the experimental section to see results on other usual benchmark datasets for UDA such as sentiment analysis + Office dataset.

Correctness: To me, the methodology adopted by the paper is correct. The method is validated on simulated data as well as real datasets and the performed measures seem relevant.

Clarity: The paper is extremely well-written and pleasant to read. Thereafter, I provide a list of various typos I could notice: - l.89: join distribution => joint distribution - l.133: "if it is known" repeated (already said at the beginning of the sentence l. 126-127) - l.134: if fact => in fact - l.211: is to Find => is to find

Relation to Prior Work: The state of the art is well-known and to my knowledge the paper positions itself clearly in this literature.

Reproducibility: Yes

Additional Feedback: Update after rebuttal: I would like to thank the authors for their detailed answers. After reading the authors' response and the other reviews, I think the following points need further discussion: theoretical validation of the proposed model, generalizability of the method (in particular to the single source case). Also, some further empirical studies could be lead. However, the overall approach proposed by the paper is novel and I think it opens news perspectives for tackling UDA. I encourage the authors to improve the points discussed in the reviews and partially answered in the response.


Review 3

Summary and Contributions: The paper presents an approach for performing unsupervised domain adaptation with possibly multiple source domains and no labeled data in the target domain. Differences in the distribution across the domains are encoded via a special type of graphical model known as an augmented graphical model that could be different than the causal graph representing the data generating process. The problem of learning a predictor in the target domain is then posed as inference on the augmented graphical model. Both synthetic and real worlds experiments show support for the proposed approach.

Strengths: The paper is easy to follow with a few minor typos that can be improved. The work addresses the issues with learning the true causal relations from observational data as well as the incapability of causal graphs to reflect the changes in distribution across domains, which is an important topic in this area. The main contribution of the work is to perform inference on the augmented graphical model in the target domain; posed as domain adaptation. The conditionals changing across the environment are modeled using latent variable conditional GANs which is an interesting application of the CGANs in order to identify the differences across the environments. Since it is intractable to obtain the conditional distribution (P(Y|X)) in the target domain, the approach utilizes a stochastic variational inference procedure to approximate the posterior P(theta|X, Y) both in the source and target domains. The empirical evaluation is sound across the synthetic and real-world settings. However, some of the real-world experiments could be moved to the supplementary material for a clearer motivation and contributions of the approach.

Weaknesses: I have some questions as well as some suggestions, clearing out them could help strengthen the existing work. - The proposed method can be thought of as a two-phase approach involving identifying and learning the augmented causal graph (including special variables such as context variables along with X and Y) and then learning a predictor in the target domain based on the invariant factors across the source and target domain as well as learning the specific distribution changes in the target domain as compared to the source domains. In relation to the first part of identifying the distribution changes across environments, I find that certain comparisons are missing that I'd like to point out. First, the idea is similar to [1] which also aims to encode the changes in the distribution across the domains using a graphical model as well. Secondly, there have been attempts at using observational data to identify the differences across domains using two-sample testing approaches [2]. Such methods can aid in understanding not just where the differences exist across domains but also analyze the effect of these shifts on the posteriors. Discussion on such methods is missing in the current setting. - The motivation for using latent-variable CGAN is not very clear. Since Gaussian Processes also are capable of approximating the posteriors, the merit of CGAN seems blur here. A better motivation for using the said approach can help clarify the said advantages. - Can multi-environment causal discovery approaches be used for learning the causal graph across all domains, which can then be combined with the domain adaptation methods? [1] Subbaswamy, Adarsh, and Suchi Saria. "I-SPEC: An End-to-End Framework for Learning Transportable, Shift-Stable Models." arXiv preprint arXiv:2002.08948 (2020). [2] Rabanser, Stephan, Stephan Günnemann, and Zachary Lipton. "Failing loudly: An empirical study of methods for detecting dataset shift." Advances in Neural Information Processing Systems. 2019.

Correctness: The claims and methods are correct.

Clarity: The paper is an easy read but should be checked for minor typos.

Relation to Prior Work: Discussion on prior work is missing in certain areas that I have listed above. Although the method is interesting, it is not easy to evaluate how it builds upon prior work since parallels across similar approaches are missing.

Reproducibility: Yes

Additional Feedback: Updates: I would like to thank the authors for their response. I have read the rebuttal and other reviewers’ comments. The authors have addressed my questions. I am of the opinion that the paper presents an interesting research direction but still needs additional theoretical grounding to ensure that the merits outweigh the flaws. While the authors have discussed the reviewer's concerns, it is still needed to evaluate guarantees even when the target data belongs to the same mother distribution but the changes are not reflected in the source datasets. Suggestions made by R1 are highly relevant and I would urge the authors to address those. I also agree with R4’s view on graph identification in high dimensional settings and appreciate the author’s comment on enhancing the method. R2’s point about accounting for changes in target distribution can help in developing guarantees. That being said, maybe by incorporating the reviewers’ suggestions into account, the authors can make a stronger case. --------------------------------------------------------------------------------------------------- The approach is interesting, however, a clear motivation for the said approach is missing that can help situate the work broader into contributions to the community. - Please refer to the suggestions mentioned above to improve the clarity as well as fill the gaps with existing work.


Review 4

Summary and Contributions: The paper provides a data-driven approach to tackle unsupervised domain adaptation problem when the source and target domains come from a common, underlying, ‘mother’ distribution. The main contribution lies in constructing the compact description of the joint distributions, by factorizing it into augmented DAGs. It is assumed that only a few factors of the joint distributions change across domains, and the graph remains the same for the distributions in different domains. With an illustrative example, the authors then propose a working algorithm for learning such augmented graphical models directly from data and inferring targets, given some target-domain features.

Strengths: The paper is well structured and the authors did a great job of explaining the problem setting and motivating their solution. I actually enjoyed reading the paper. The illustrative 'toy' graph in Figure 1, helped me understand the framework better. Moreover, I liked that the authors do not assume the underlying graph to be known and use some previously proposed techniques to first find the augmented DAGs and then use it for their end-to-end domain adaptation framework. The empirical results look very good and make a strong case for accepting this work.

Weaknesses: I do not have any major issues with the paper. Though I do not believe that the following questions are weaknesses, I would appreciate it if the authors could answer them. 1. How much domain adaptation results rely on learning the correct augmented DAG? For e.g. if the identified DAG is wrong then how much that affects the DA results. 2. This might not be relevant, but could authors comment on how would they enhance this method when the DAG is not reasonably identifiable for e.g. in high-dim data (images) when it's hard to factorize the data (commonly posed as disentanglement). Could there be a way to encode uncertainty in the structure of the graph to accommodate this?

Correctness: The presented method and empirical results seem correct. The authors provided code, I briefly went through it and found it to be okay. I must admit, I do not know some of the relevant literature referred to in the paper. I also didn't thoroughly check the derivations in the appendix.

Clarity: Yes, it is a well-written paper. Some minor typos in line 72, 102, 134-135.

Relation to Prior Work: I think the paper cites the prior work comprehensively. I may have missed some references though.

Reproducibility: Yes

Additional Feedback: Kindly address the questions in weaknesses section. ------------------------------------------------------------- Update: I thank the authors for addressing my comments. However, after reading other reviewers’ comments and the rebuttal, I believe that the work can be improved further. I agree with R1, R2, and R3 that the discussion on the connections between source and target domains is a little obscure. Given the context of this work, it’s an important issue. Either theoretical guarantees or empirical validations (even on synthetic example), addressing this aspect, would significantly improve the paper. I have likewise changed my score.

[Author Response · NeurIPS 2020]

We sincerely thank all reviewers for their time and helpful comments. Below please see our response to the comments.

**Q1 (R1).** "what kind of guarantees," & "relations between domains and assumptions on test data." The assumptions was given in lines 47-52: domain-specific distributions are random draws from a "mother" distribution. We have asymptotic guarantees (as the number of domains and sample size in each domain increases). With a graphical representation to describe essential properties of the mother distribution (assumed in lines 105-106), domain adaptation (DA) reduces to inference on the graphical model, where all relevant information for the label in the target domain is exploited.

**Q2 (R1)** "successful if the target data distribution is a convex combination of source distributions?" Yes, this is a special case satisfying our assumption. However, in complex situations, one may need to use supernodes for sets of the variables, because of the independent change assumption in the graphical representation (lines 109 & 120-124).

**Q3 (R1)** what if changes in new data are broader Nice point. It violates our assumption in lines 47-52, and our method may fail. This situation is like extrapolation, and additional assumptions may be needed. Will discuss it.

**Q4 (R1)** "causality," "interventions," and "negative transfer." Interesting point. The connection between interventions and distribution changes has been discussed in the causality community; see, e.g., "Interventions and causal inference" (by Eberhardt and Scheines, 2007). Even if the graphical model is a causal representation, our technical assumptions are still needed to ensure successful transfer (please also refer to **Q3**).

**Q5 (R1&R2&R4)** "synthetic tailored to the assumptions" & "image example with two variables." Synthetic experiments aim to verify the validity of the proposal. Inevitably, we need to consider image classification, an important problem in DA, in which it is natural to consider image pixels together as a supernode (L122-124). We use $Y \rightarrow X$ in light of previous work, which is consistent with the results of our procedure, since $P(Y)$ does not change in the image datasets.

**Q6 (R1)** single-source case. For instance, if we know $P(Y)$ does not change (as an additional constraint), then we know the structure and can do adaptation. Without such constraints, multiple domains would be needed.

**Q7 (R1&R4)** Influence of errors in graph learning. Thanks for raising this practical issue. We have conducted quantitative analysis following your suggestions and found that the adaptation performance indeed becomes worse when the result of the first step is wrong. The results will be included in the paper. For instance, when the direction between $X_2$ and $X_3$ in the graph is wrong, the prediction accuracy drops to $92(6.3)$, $82.36(10.33)$, and $83.1(9.95)$, with 2, 4, and 9 source domains, respectively.

**Q8 (R1)** results in Table 1. As confirmed by additional experiments, it is because of randomness in the generated datasets. we keep the target domains to be those in the 9 source experiment, and randomly sample 2/4/9 source domains. The results of our method on 2, 4, and 9 source domains are $81.04(12.71)$, $82.88(10.31)$, and $83.90(9.02)$. However, the comparisons of the methods are still fair because they deal with the same datasets.

**Q9 (R2&R1)** computational complexity. Our procedure is essentially local graph learning (focusing on only $Y$ and variables in its Markov blanket) and inference on graphical models. We will include the complexity for both modules. The complexity is not very sensitive to the original dimensions of the data, but to how large the Markov blanket is. Empirically, it took around 6 hours on the Wifi data with 69 variables. The LVGAN training is efficient on GPUs. It takes about 20 mins on the WiFi data and 60 mins on the digits data.

**Q10 (R2))** learning a graphical model over images. Please refer to **Q5**.

**Q11 (R2)** Different causal structure graph learning on simulated data. We appreciate your insight and valuable suggestions. We have done additional experiments regarding this point, please refer to **Q7**. Following your suggestions, we will include the results and discussions in the paper. We applied the whole procedure to learn the graph and do inference. We will also include the performance of graph learning in the paper.

**Q12 (R2)** visualize relevant features in real datasets. We have shown the graph learned on the Wifi data in the Appendix A6. However, we consider image pixels together as a supernode.

**Q13 (R2)** office and sentiment. Thanks for your suggestion. We are playing with the datasets you recommended.

**Q14 (R3)** relevant works [1][2]. Many thanks for recommending the recent works. We will cite them and discuss how they are connected to our work. [1] uses the source domains to find invariant features and use them to predict in the target domain, and is closely related, although we make use of both invariant and changing features. The two-sample test method [2] is actually a special case of our conditional independence test-based method, because the dependence between a continuous variable (e.g. X) and discrete variable (domain index C) is related to the discrepancy between $P(X|C = 1)$ and $P(X|C = 0)$. At the same time, we hope that the reviewer kindly understands why the two recent works were not discussed: the first was available on arxiv this year and the other was published at last year's NeurIPS.

**Q15 (R3)** motivation of latent-variable GAN. We develop the latent-variable GAN model because it is an implicit model that is flexible in modeling distributions, especially for images. We agree that other methods, including GP-based ones, may work well in various scenarios. Will make it explicit in the paper.

**Q16 (R3)** Multi-environment causal discovery. Yes. In fact, our augmented graph learning method is extended from a recent multi-environment causal discovery method CD-NOD. The details are given in Sec 3.1 and Appendix A3.

**Q17 (R4)** DA performance on DAG. Thanks for your encouraging comments. Please refer to **Q7**.

**Q18 (R4)** Enhance when DAG is not identifiable in images. Great idea! Learning a graph in hidden representation of images can further enhance our method.

[Meta-Review · NeurIPS 2020]

This paper presents a novel framework for unsupervised domain adaptation. Overall reviewers and meta-reviewer appreciate the ideas of a ‘mother distribution’ but also point out that there are limitations since changes in source domains may not be reflected in target domains. A discussion on the soundness of the method under specific generative/causal model assumptions would be very useful and is strongly encouraged. Modeling conditional distributions via CGAN as proposed in Sec. 3.2 has first been proposed in CausalGAN by Kocaoglu et al. in the context of learning causal models, except for the additional theta parameter. We encourage the authors to discuss this relation.